# Regio- and Stereoselective Switchable Synthesis of (*E*)- and (*Z*)-*N*-Carbonylvinylated Pyrazoles

**DOI:** 10.3390/molecules28114347

**Published:** 2023-05-25

**Authors:** Xue Zhang, Zheyu Zhang, Haifeng Yu, Guangbo Che

**Affiliations:** College of Chemistry, Baicheng Normal University, Baicheng 137000, China; zhangx09236@163.com (X.Z.); zheyu_z@163.com (Z.Z.)

**Keywords:** silver carbonate, Michael addition, pyrazoles, alkynes, stereoselectivity

## Abstract

Regio- and stereoselective switchable synthesis of (*E*)- and (*Z*)-*N*-carbonylvinylated pyrazoles is first developed by using the Michael addition reaction of pyrazoles and conjugated carbonyl alkynes. Ag_2_CO_3_ plays a key role in the switchable synthesis of (*E*)- and (*Z*)-*N*-carbonylvinylated pyrazoles. Ag_2_CO_3_-free reactions lead to thermodynamically stable (*E*)-*N*-carbonylvinylated pyrazoles in excellent yields whereas reactions with Ag_2_CO_3_ give (*Z*)-*N*-carbonylvinylated pyrazoles in good yields. It is noteworthy that (*E*)- or (*Z*)-*N*^1^-carbonylvinylated pyrazoles are obtained with high regioselectivity when asymmetrically substituted pyrazoles react with conjugated carbonyl alkynes. The method can also extend to the gram scale. A plausible mechanism is proposed on the basis of the detailed studies, wherein Ag^+^ acts as coordination guidance.

## 1. Introduction

*N*-Functionalized pyrazoles are often found in many products with high biological and pharmacological activities [1,2,3,4,5,6,7,8]. During the past two decades, the synthesis of *N*-vinylated pyrazoles as an important subset of *N*-functionalized pyrazoles has attracted huge interest from chemists due to their privileged core structures in many biologically active products (Figure 1) [9,10,11,12] and the good further transformations of their alkenyl group in organic synthesis [13,14,15]; therefore, various synthetic methods have been developed. These methods include the La(OTf)_3_-catalyzed tandem aza-Michael addition/cyclization reactions of *N*-tosylhydrazones and alkynes [16], Et_3_N-mediated *N*-vinylation of (β-trifluoromethyl)vinyl sulfonium salt and pyrazoles [17], copper-catalyzed *N*-vinylation of pyrazoles and vinyl halides or vinylboronic acids [18,19,20,21,22], photocatalytic dehydrogenative cross-coupling of pyrazoles with electron-poor alkenes [23], and the Lewis-acid- or Ruthenium-complex-catalyzed addition of pyrazoles to alkynes [24,25]. However, these reported methods have usually produced thermodynamically stable (*E*)-*N*-vinylated pyrazoles or a mixture of *E*/*Z* isomers. In fact, the *Z* and *E* isomers of products usually exhibit different biological activities. For example, (*Z*)-1-ethyl-7-((4-methoxy-3-methylphenyl)diazenyl)-4-oxo-1,4-dihydroquinoline-3-carboxylic acid was found to be 4- and 8-fold active on *E*. *coli* (Gram-negative) and *M. luteus* (Gram-positive) [26,27]. Therefore, it was necessary and highly desirable to develop the switchable synthesis of (*Z*)- and (*E*)-*N*-vinylated pyrazoles in regio- and stereoselective fashions. Recently, several research groups have focused on this investigation. In 2010, Cook and co-workers stereoselectively prepared (*Z*)- and (*E*)-*N*-vinylated pyrazoles by copper-catalyzed cross-coupling reactions of pyrazoles with (*Z*)- and (*E*)-vinyl halides (Figure 1A) [28]. From the viewpoint of green chemistry, this method is not atomically economical and environmentally friendly due to it producing large amounts of alkali halide and using expansive vinyl halides. Later, to develop synthetic methods with atomic economy, based on the strategy of catalyst-controlled product stereoselectivity, Bhattacharjee and co-workers developed Ru complex- (i.e., [Ru(dppe)(PPh_3_)(CH_3_CN)_2_Cl][BPh_4_] and [Ru(dppp)_2_(CH_3_CN)Cl][BPh_4_])-catalyzed stereoselective *N*-vinylation of pyrazoles and alkynes, and efficiently synthesized (*E*)- and (*Z*)-*N*-vinylated pyrazoles, respectively (Figure 1B) [29]. In another recent report, Verma and co-workers realized KOH-mediated stereoselective synthesis of (*Z*)- and (*E*)-*N*-vinylated pyrazoles by the addition of pyrazoles to alkynes, in which the stereochemical outcome of the reaction was governed by time and the quantity of the base (Figure 1C) [30]. Although two examples revealed good stereoselective synthesis of (*Z*)- and (*E*)-*N*-vinylated pyrazoles by the addition of pyrazoles to alkynes, no work has been directed to study the stereoselective addition of pyrazoles with conjugated carbonyl alkynes to prepare *N*-carbonylvinylated pyrazoles, presumably due to the incompatibility of carbonyl group with its optimal reaction conditions. In addition, in their work, no in-depth and detailed investigations into the regio- and stereoselective *N*^1^ or *N*^2^ vinylation of the asymmetrically substituted pyrazoles and alkynes were performed. So, the regio- and stereoselective switchable synthesis of (*E*)- and (*Z*)-*N*-carbonylvinylated pyrazoles is challenging.

Considering the importance of pyrazoles, our recent research interests have focused on the synthesis of pyrazoles. A variety of *N*-unsubstituted pyrazoles bearing indole units have been synthesized from the cyclocondensation reaction of β-ethyltho-β-indolyl-α, β-unsaturated ketones and hydrazine hydrate or semicarbazide hydrochlorides as a hydrazine equivalent in organic solvent or water [31,32,33]. Furthermore, by controlling the reaction conditions, the regioselective synthesis of isomeric 3-(1-substituted pyrazol-3(5)-yl) indoles was also successfully realized when β- ethyltho-β-indolyl-α, β-unsaturated ketones reacted with monosubstituted hydrazines [34]. Subsequently, we developed the Ag_2_CO_3_-catalyzed aza-Michael addition of pyrazoles to α, β- unsaturated carbonyl compounds, to afford a series of *N*-alkylated pyrazoles in excellent yields and with high regioselectivity [35]. As a continuation of our interest in the synthesis of pyrazole derivatives, most recently, we studied the regio- and stereoselective aza-Michael addition of pyrazoles and conjugated carbonyl alkynes for the switchable synthesis of (*E*)- and (*Z*)-*N*-carbonylvinylated pyrazoles (Figure 1D). It was found that pyrazoles efficiently reacted with the conjugated carbonyl alkynes in the absence of Ag_2_CO_3_ to preferentially generate (*E*)-*N*-carbonylvinylated pyrazoles in excellent yields, while in the presence of 50 mol% of Ag_2_CO_3_, the reactions favored to afford (*Z*)-*N*-carbonylvinylated pyrazoles in good yields, in which Ag_2_CO_3_ played the key role. Herein, we report our finding.

## 2. Results and Discussion

The reaction of 3, 5-dimethyl-1*H*-pyrazole **1aa** and ethyl propiolate **2a** was chosen to screen the reaction conditions. The summary of these results is listed in Table 1. We proceeded the first reaction without any additives in 1, 2-dichloroethane (DCE) at room temperature for 24 h, and found this reaction produced two stable products in 60% and 30% yields, respectively (Table 1, entry 1). The products were characterized as (*E*)-ethyl 3-(3, 5-dimethyl-1*H*-pyrazol-1-yl) acrylate **3a** (60%) and its (*Z*)-isomer **4a** (30%) from the spectral and analytical data, which indicated that the reaction gave (*E*)-**3a** in preference to (*Z*)-**4a**. The reaction showed dependence on the reaction temperature; with increasing reaction temperature, the reaction became more efficient, in which the reaction time was significantly shorter and the yield of (*E*)-**3a** was markedly higher (Table 1, entries 2–5), and (*E*)-**3a** was obtained at 90% yield when the reaction ran at 60 °C for 8 h (Table 1, entry 4). Using other solvents such as toluene and 1, 4-dioxane, the reaction efficiency was not significantly improved (Table 1, entries 6 and 7). Because Ag^+^ easily coordinates with both the nitrogen atom of imine and oxygen atom of the carbonyl group [35,36,37,38], we envisioned that the coordination guidance of Ag^+^ probably favors the formation of (*Z*)-isomer **4a.** Thus, with 10 mol% of readily available Ag_2_CO_3_ as the catalyst, we proceeded the reaction in DCE at 60 °C and 20 °C, respectively. As expected, **4a** as the major product was generated at 51% and 29% yields, respectively (Table 1, entries 8 and 9). Encouraged by this result, we further increased the amount of Ag_2_CO_3_ (Table 1, entries 10–15). Raising the amount of Ag_2_CO_3_ to 50 mol% effectively improved the yield of **4a** to 81% (Table 1, entry 13), while when further raising the amount of Ag_2_CO_3_, the yield of **4a** failed to be obviously improved (Table 1, entries 14 and 15). Of the various reaction temperatures checked, 60 °C was found to be optimal (Table 1, entries 16 and 17). The use of AgOAc and AgNO_3_ as the catalyst led to the low formation of **4a** (Table 1, entries 18 and 19). Accordingly, the reaction conditions were optimized as follows: condition A for the synthesis of **3** (Table 1, entry 4): DCE as solvent and 60 °C; condition B for the synthesis of **4** (Table 1, entry 12): DCE as solvent, 50 mol% of Ag_2_CO_3_ as catalyst and 60 °C.

After optimizing the reaction conditions, we explored the universality of the procedure (Figure 2). Firstly, the generality of the reactions of various symmetrically substituted pyrazoles and conjugated carbonyl alkynes was examined, and the results are summarized in Figure 2. Using ethyl propionate **2a** as the partner, we initially studied the scope of symmetrically substituted pyrazoles **1a**. A variety of **1a**, such as 3, 5-identically disubstituted pyrazoles **1aa**–**1ad**, 4-substituted pyrazoles **1ae**–**1ao**, and 3, 5-identical disubstituted-4-substituted pyrazoles **1ap**–**1ar**, smoothly reacted with **2a** under the optimized conditions A and B to efficiently afford corresponding *N*-carbonylvinylated pyrazoles **3a**–**3r** and **4a**–**4r** in good yield and with good stereoselectivity, respectively. Clearly, the steric effects and electronic characters of substituents in **1a** revealed a significant effect on the reaction rate. With the increase in steric bulk of the C3 or C5 substituent of **1aa**–**1ad**, the reaction rate decreases. Pyrazoles with electron-donating groups, such as Me (**1af**), OCH_3_ (**1ag**) and Ph (**1ah**), displayed faster reaction rates than pyrazoles with electron-withdrawing groups, such as X (**1ai**–**1ak**), NO_2_ (**1al**), CN (**1am**), CO_2_Et (**1an**), and COCH_3_ (**1ao**). Moreover, many active substituents, such as X, NO_2_, CN, CO_2_Et, and COCH_3_, of **1a** could be well tolerated in the chemical transformations. Then, under the optimized reaction conditions A and B, the scope of conjugated carbonyl alkynes **2** was examined. Like **2a**, methyl propionate **2b**, isopropyl propionate **2c** and tert-butyl propiolate **2d** efficiently reacted with **1aa** to produce *N*-carbonylvinylated pyrazoles **3s**–**3u** and **4s**–**4u** at good yields and with high stereoselectivity, respectively. In the case of 1-phenylprop-2-yn-1-one **2e**, the desired products **3v** and **4v** were also obtained at 92% yield and 87% yield, respectively.

Next, under the optimal reaction conditions A and B, we investigated in detail the reactions of the asymmetrically substituted pyrazoles **1b** with conjugated carbonyl alkynes **2** to examine the regioselectivity of the method. When 3-substituted pyrazoles **1ba**–**1bh** reacted with **2a**, *N*^1^-carbonylvinylated pyrazoles **5a**–**5h** and **6a**–**6h** were always obtained as major products with good regioselectivity, respectively, and, especially, **1bg** exclusively afforded **5g** (82% yield) and **6g** (84% yield). Although the electronic character imparted an obvious impact on the reaction rates, they had no significant influence on the regioselectivity of the reactions. Notably, since R_f_ values of **5h** and **5h′** are the same, their mixtures were obtained when the reaction of **1bh** and **2a** was carried out under condition A, in which their molar ratio was determined as 2:1 from ^1^H NMR analysis. The compound 3-Methyl-5-phenyl-1*H*-pyrazole **1i** also favored the formation of **5i** with a **5i**/**5i′** ratio of 2:1 and **6i** with a **6i/6i’**ratio of 2:1, suggesting that the steric effect of the C5 substituent of **1i** seemingly exerted a negligible influence on the regioselectivity. In the same fashion, a variety of conjugated carbonyl alkynes **2b**–**2e** also efficiently reacted with **1ba** to afford **5j**–**5m** and **6j**–**6m** with good regioselectivity. Except for **5h** and **5h′**, the other isomeric products of **5** and **5′**, or **6** and **6′** could easily be isolated by column chromatography due to their different R_f_ values. It is noteworthy that the molecular structures of **3l**, **4o**, **5c**, **5k′** and **6c** were further confirmed by X-ray crystallographic analysis (Figure 2) [39]. In **3l**, **5c** and **5k′**, the pyrazolyl and carbonyl groups are positioned *anti* to each other (*E* configuration), and these two groups are positioned *cis* to each other in **4o** and **6c** (*Z* configuration).

Furthermore, we carried out gram-scale reactions to demonstrate the practicality of the chemical procedure (Figure 3 and Figure 4). Gratifyingly, **1aa** (0.96 g, 10.0 mmol) effectively reacted with **2a** (1.18 mL, 12.0 mmol) to successfully afford 1.57 g (81% yield) of **3a** and 1.455 g (75% yield) of **4a** under the optimized conditions A and B.

Based on the above results, a plausible mechanistic pathway for the formation of **3** and **4** was proposed (Figure 5). In the absence of Ag_2_CO_3_, **1aa** and **2a** underwent a Michael addition reaction to directly give (*E*)-**3a** due to thermodynamical stability of the *E* configuration (Path a). In the presence of Ag_2_CO_3,_ the reaction proceeded based on Path b. Since Ag^+^ easily coordinates with both the nitrogen atom of imine and oxygen atom of the carbonyl group [35,36,37,38], it is clear that the reaction commenced from the coordination of Ag^+^ to *N*^2^ of **1aa** and carbonyl of **2a** to form intermediate **I**, in which **1aa** and **2a** are close enough and **2a** is also fully activated. Subsequently, **I** occurs Michael addition to produce the intermediate **II,** in which pyrazolyl and acyl groups are positioned *cis* to each other by the coordination of Ag^+^. Finally, in the presence of AgCO_3_^−^, intermediate **II** produces (*Z*)-**4a**. The generated Ag_2_CO_3_ further completes the catalytic cycle.

## 3. Materials and Methods

### 3.1. General Information

^1^H and ^13^C{1H} NMR spectra were recorded on a Bruker DRX-600 spectrometer and all chemical shift values referred to δ TMS = 0.00 ppm (δ (^1^H)) and CDCl_3_(δ (^13^C), 77.16 ppm). The HRMS analysis was achieved on a Bruck microTof by using the ESI method. All the melting points were uncorrected. Analytical TLC plates with Sigma-Aldrich silica gel 60F200 were viewed by UV light (254 nm). Chromatographic purifications were performed on SDZF silica gel 160 (Appendix A).

### 3.2. Typical Procedure for the Synthesis of **3a**

A solution of 3,5-dimethyl-1*H*-pyrazole (**1aa**) (24.0 mg, 0.25 mmol) and ethyl propionate (**2a**) (30.4 μL, 0.3 mmol) was reacted in DCE (1 mL) at 60 °C. After the reaction was finished as indicated by TLC (reaction time, 8 h), the resulting mixture was poured into water (10 mL) and extracted with DCM (CH_2_Cl_2_, 3 × 10 mL). The combined organic layer was dried over anhydrous Na_2_SO_4_ and concentrated in vacuo. Purification of the crude product with flash column chromatography (petroleum ether (60–90 °C)/ethyl acetate = 10:1, *v*/*v*) was carried out to give 3a (43.7 mg, 90%).

### 3.3. Typical Procedure for the Synthesis of **4a**

Ag_2_CO_3_ (34.5 mg, 0.125 mmol) was added to a solution of 3,5-dimethyl-1*H*-pyrazole (**1aa**) (24.0 mg, 0.25 mmol) and ethyl propionate (**2a**) (30.4 μL, 0.3 mmol) in DCE (1 mL) at 60 °C. After the reaction was finished as indicated by TLC (reaction time, 12 h), the resulting mixture was poured into water (10 mL) and extracted with DCM (CH_2_Cl_2_, 10 mL × 3). The combined organic layer was dried over anhydrous Na_2_SO_4_ and concentrated in vacuo. Purification of the crude product with flash column chromatography (petroleum ether (60–90 °C)/ethyl acetate = 8:1, *v*/*v*) was carried out to give 4a (39.3 mg, 81%).

### 3.4. Typical Procedure for the Synthesis of **5a** and **5a’**

A solution of 3-phenyl-1*H*-pyrazole (**1ba**) (72.0 mg, 0.5 mmol) and ethyl propionate (**2a**) (101.5 μL, 1.0 mmol) in DCE (1 mL) was reacted at 60 °C. After the reaction was finished as indicated by TLC (reaction time, 8 h), the resulting mixture was poured into water (10 mL) and extracted with DCM (CH_2_Cl_2_, 3 × 10 mL). The combined organic layer was dried over anhydrous Na_2_SO_4_ and concentrated *in vacuo*. Purification of the crude product with flash column chromatography (petroleum ether (60–90 °C)/ethyl acetate = 10:1, *v*/*v*) was performed to give **5a** (79.8 mg, 66%) and **5a′** (24.2 mg, 20%).

### 3.5. Typical Procedure for the Synthesis of **6a** and **6a’**

Ag_2_CO_3_ (69.0 mg, 0.25 mmol) was added to a solution of 3-phenyl-1*H*-pyrazole (**1ba**) (72.0 mg, 0.5 mmol) and ethyl propionate (**2a**) (101.5 μL, 1.0 mmol) in DCE (1 mL) at 60 °C. After the reaction was finished as indicated by TLC (reaction time, 12 h), the resulting mixture was poured into water (10 mL) and extracted with DCM (CH_2_Cl_2_, 3 × 10 mL). The combined organic layer was dried over anhydrous Na_2_SO_4_ and concentrated in vacuo. Purification of the crude product with flash column chromatography (petroleum ether (60–90 °C)/ethyl acetate = 8:1, *v*/*v*) was performed to give **6a** (86.0 mg, 71%) and **6a′** (21.8 mg, 18%).

## 4. Conclusions

In summary, for the first time, the regio- and stereoselective synthesis of (*E*)- and (*Z*)-*N*-carbonylvinylated pyrazoles was successfully realized by the Michael addition reaction of pyrazoles and conjugated carbonyl alkynes. Symmetrically substituted pyrazoles stereoselectively gave (*E*)- and (*Z*)-*N*-carbonylvinylated pyrazoles in good yields, respectively. In the case of asymmetrically substituted pyrazoles, the reaction revealed not only excellent stereo-selectivity but also good regioselectivity. These features, such as the high regio- and stereoselectivity, commercially available catalyst, good substrate scope, mild conditions and scalability, made the methodology very practical and attractive. Further investigations on the applications of *N*-carbonyl vinylated pyrazoles are currently underway in our laboratory.

## Data Availability

Not applicable.

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
