# Peer review of "Regio- and Stereoselective Switchable Synthesis of (E)- and (Z)-N-Carbonylvinylated Pyrazoles"

_molecules, 2023, doi:10.3390/molecules28114347_

Round 1
Reviewer 1 Report
This is interesting but not very well written paper that could be of interest for readers of organic chemistry. The manuscript needs minor revision before it is considered for publication:
1. The authors used term carbonyl alkynes but products are named N-acylvinylated pyrazoles. The term acylvinyl is not correct (acyl?) and should be changed to e.g., 3-acrylic derivatives in all text except for compounds derived from 2e.
2. Please comment on X-ray crystallographic structures of Z- and E- compounds studied.
line 9 - Ag2CO3-free reaction
line 10 – reaction with Ag2CO3
line 56 – improtant should be important.
line 70 – acyl-vinylated should be acylvinylated
line 74 – numeration of the pyrazole ring should be given.
Extensive editing of English language and style is required.
Author Response
Dear Prof. Mabel Calvert
Thank you very much for the comments and suggestions about our manuscript (molecules-2395418) made by you and reviewer 1.
According the comments, we have carefully made revision on the manuscript. In the revised manuscript, all additions and modifications had been marked in red. Below is the response to you and reviewer 1 by point to point:
This is interesting but not very well written paper that could be of interest for readers of organic chemistry. The manuscript needs minor revision before it is considered for publication:
- The authors used term carbonyl alkynes but products are named N-acylvinylated pyrazoles. The term acylvinyl is not correct (acyl?) and should be changed to e.g., 3-acrylic derivatives in all text except for compounds derived from 2e.
Response: The suggestions are very good, and after careful studying, we have revised “N-acylvinylated pyrazoles” to “N-carbonylvinylated pyrazoles”.
- Please comment on X-ray crystallographic structures of Z- and E- compounds studied.
Response: We have added relevant comments in the results and discussion section of the revised manuscript (i.e, it is noteworthy that the molecular structures of 3l, 4o, 5c, 5k’ and 6c were further confirmed by X-ray crystallographic analysis (Fig. 2) . [39] In 3l, 5c and 5k’, the pyrazolyl and carbonyl groups are positioned anti to each other (E configuration), and these two groups are positioned cis to each other in 4o and 6c (Z configuration)).
line 9 - Ag2CO3-free reaction
Response: We had corrected it in the revised manuscript.
line 10 – reaction with Ag2CO3
Response: We had corrected it in the revised manuscript.
line 56 – improtant should be important.
Response: We had corrected it in the revised manuscript.
line 70 – acyl-vinylated should be acylvinylated
Response: We had corrected it in the revised manuscript.
line 74 – numeration of the pyrazole ring should be given.
Response: We had corrected it in the revised manuscript (see scheme 1, D this work).
In addition, we have invited experts with good English to revise this manuscript, and English language and presentation are more easy and smooth.
Finally, we want to show our great respect to you and reviewers. Your critical reviews and suggestions definitely improved the quality of this manuscript.
Thank you once again for your efforts in this matter. I look forward to hearing from you.
Sincerely yours,
Haifeng Yu

Reviewer 2 Report
The authors showed that the presence/absence of Ag2CO3 can switch the (E)/(Z) stereoselectivity of N-acylvinylated pyrazoles (22 examples). Further, in the case of asymmetrically substituted pyrazoles, the presence/absence of Ag2CO3 also directs the stereoselectivity but also led to high regioselectivity (13 examples). In general, the manuscript has reliable results, is well-written and compounds are properly characterized.
Overall, this reviewer recommends being published on Molecules. Though, some concerns should be addressed before acceptance.
Page1, Line 32. Correct E. coli
Page3, Line83-85. A comment about the effect of temperature on the time (h)?
Page3, Line 94, 95 and 97. Explain “…the yield of 4a failed to be obviously improved” (?). Reformulate “60 oC seemed.. ”. Low yields afforded with AgOAc or AgNO3 were because of less molarity of Ag (I) ion or because of differences on basicity?
Page 4, Lines 129-145. Pairs 5x-5x’ or 6x-6x’ products have similar H- and C- spectra (profile) since they are N-isomers. It could be very very helpful to include in the body of the manuscript a couple of examples -one for 5 / 5’ and another for 6 / 6’- to complement X-ray analysis shown in Fig2. NOESY or HMBC spectra can clarify the correct assignment of regioisomers.
Page 6, Scheme 3. Alkyne from structure I should be corrected.
Page 6, Line 189 and 192. Please double check time and yield. Shouldn’t be in accordance with Entry12 Table1?.
Please, double check inside SI:
Page S5. Compounds 3i, 3j and 3k. each aromatic signal is a singlet. Compound 3k, signal from one carbon is missing near 60.94 ppm (quaternary C linked to -I?). Compound 3l, instead of 137,2, should be 138.1ppm.
Page S8. Compound 3t. Signals 1.27 and 1.29ppm shouldn’t be d, instead of s?.
Author Response
Dear Prof. Mabel Calvert
Thank you very much for the comments and suggestions about our manuscript (molecules-2395418) made by you and reviewer 2.
According the comments, we have carefully made revision on the manuscript. In the revised manuscript, all additions and modifications had been marked in red. Below is the response to you and reviewer 2 by point to point:
The authors showed that the presence/absence of Ag2CO3 can switch the (E)/(Z) stereoselectivity of N-acylvinylated pyrazoles (22 examples). Further, in the case of asymmetrically substituted pyrazoles, the presence/absence of Ag2CO3 also directs the stereoselectivity but also led to high regioselectivity (13 examples). In general, the manuscript has reliable results, is well-written and compounds are properly characterized.
Overall, this reviewer recommends being published on Molecules. Though, some concerns should be addressed before acceptance.
- Page1, Line 32. Correct E. coli
Response: We had corrected it in the revised manuscript.
- Page3, Line83-85. A comment about the effect of temperature on the time (h)?
Response: We have added the comments in the revised manuscript (i.e., The reaction showed dependence on the reaction temperature, with increasing the reaction temperature, the reaction became more efficient, in which the reaction time was significantly shorter and the yield of (E)-3a was markedly higher).
- Page3, Line 94, 95 and 97. Explain “…the yield of 4afailed to be obviously improved” (?). Reformulate “60 oC seemed.. ”. Low yields afforded with AgOAc or AgNO3 were because of less molarity of Ag (I) ion or because of differences on basicity?
- Explain “…the yield of 4afailed to be obviously improved” (?)
Response: In our study, we found that increasing the amount of Ag2CO3 or prolonging the reaction time resulted in a decrease in the yield of 4a. However, to explain this phenomenon, we have consulted a large number of relevant literature, but no corresponding explanations are provided.
- Reformulate “60 oC seemed.. ”
Response: In the revised manuscript, we have revised this statement(i.e., Of the various reaction temperature checked, 60 oC was found to be optimal).
- Low yields afforded with AgOAc or AgNO3were because of less molarity of Ag (I) ion or because of differences on basicity?
Response: This reason is not very clear, and may be related to the solubility of silver salts, the properties of acid ions, etc., which requires further research by us.
- Page 4, Lines 129-145. Pairs 5x-5x’ or 6x-6x’ products have similar H- and C- spectra (profile) since they are N-isomers. It could be very very helpful to include in the body of the manuscript a couple of examples -one for 5 / 5’ and another for 6 / 6’- to complement X-ray analysis shown in Fig2. NOESY or HMBC spectra can clarify the correct assignment of regioisomers.
Response: In the revised manuscript, we added comments on single crystal analysis, in which the stereoisomerism and regional isomerism of the product were well explained, so we did not supplement NOESY and HMBC spectral analysis.
- Page 6, Scheme 3. Alkyne from structure Ishould be corrected.
Response: We had corrected it in the revised manuscript.
- Page 6, Line 189 and 192. Please double check time and yield. Shouldn’t be in accordance with Entry12 Table1?.
Response: We had corrected it in the revised manuscript.
- Please, double check inside SI:
Page S5.
- Compounds 3i, 3j and 3k. each aromatic signal is a singlet.
Response: Compounds 3i, 3j and 3k. each aromatic signal should be a singlet, and we had corrected their spectrogram and spectra analysis in the revised SI.
- Compound 3k, signal from one carbon is missing near 60.94 ppm (quaternary C linked to -I?).
Response: We have added its signal from one carbon in 61.18 ppm in the revised SI.
- Compound 3l, instead of 137,2, should be 138.1ppm.
Response: We had corrected it in the revised SI.
- Page S8. Compound 3t. Signals 1.27 and 1.29ppm shouldn’t be d, instead of s?.
Response: We had corrected it in the revised SI.
In addition, we have invited experts with good English to revise this manuscript, and English language and presentation are more easy and smooth.
Finally, we want to show our great respect to you and reviewers. Your critical reviews and suggestions definitely improved the quality of this manuscript.
Thank you once again for your efforts in this matter. I look forward to hearing from you.
Sincerely yours,
Haifeng Yu

Reviewer 3 Report
Yu, Che, and co-workers developed a highly selective method to synthesis (E)- and (Z)-N-acylvinylated pyrazoles separately by varying the amount of Lewis acid silver carbonate. Although several groups have contributed to the study of the stereoselective preparation of (Z)- and (E)-N-vinylated pyrazoles, the authors successfully extended the substrate scope to conjugated carbonyl alkynes. The procedure is simple and the reaction condition is mild. This work is comprehensive, and the manuscript is very well-organized. I recommend its publication in Molecules after addressing the following aspects.
1. In Table 1, the authors tested up to 60 mol% of Ag2CO3, have the authors tried to further increase the amount to see if the ratio increases or not?
2. In Table 1, the authors ran the Ag2CO3 mediated reaction under 60 degree Celsius, have the authors tried to lower the reaction temperature to see if the product ratio is still good or increased? Right now, the 60% Ag2CO3 is not a catalytic amount, have authors tried to lower the amount of Ag2CO3 under low reaction temperature (Such as room temperature or lower)?
Moderate editing of the English language is required. For example:
i) In line 41-43, “Bhattacharjee and co-workers developed Ru complexes with different structures (i. e [Ru(dppe)(PPh3)(CH3CN)2Cl][BPh4] and [Ru(dppp)2 (CH3CN)Cl][BPh4]) catalyzed stereoselective N-vinylation of pyrazoles and alkynes…” Remove “with different structures”.
ii) In line 57-58, “In consideration of the importance of pyrazoles, we recently were interested in the synthesis of pyrazoles.” can be modified to “Considering the importance of pyrazoles, our recent research interests focused on the synthesis of pyrazoles.”
iii) In line 83-84, “The reaction showed dependence on the reaction temperature, becoming more efficient with an increase in reaction temperature..” can be modified to “The reaction showed dependence on the reaction temperature, with increasing the reaction temperature, the reaction became more efficient.”
iv) In line 91-92, “Encouraged by this result, with the aim of increasing the yield of 4a, we studied the reaction in detail” can be modified to “Encouraged by this result, we further increased the amount of Ag2CO3.”
v) In line 95-96, “60oC seemed to be a suitable reaction temperature ofter the reaction temperature was checked” needs to be rewritten.
Author Response
Dear Prof. Mabel Calvert
Thank you very much for the comments and suggestions about our manuscript (molecules-2395418) made by you and reviewer 3.
According the comments, we have carefully made revision on the manuscript. In the revised manuscript, all additions and modifications had been marked in red. Below is the response to you and reviewer 3 by point to point:
- Response to Reviewer 3's comments and suggestions
Yu, Che, and co-workers developed a highly selective method to synthesis (E)- and (Z)-N-acylvinylated pyrazoles separately by varying the amount of Lewis acid silver carbonate. Although several groups have contributed to the study of the stereoselective preparation of (Z)- and (E)-N-vinylated pyrazoles, the authors successfully extended the substrate scope to conjugated carbonyl alkynes. The procedure is simple and the reaction condition is mild. This work is comprehensive, and the manuscript is very well-organized. I recommend its publication in Molecules after addressing the following aspects.
- In Table 1, the authors tested up to 60 mol% of Ag2CO3, have the authors tried to further increase the amount to see if the ratio increases or not?
Response: When further increasing the amount of Ag2CO3, although the molar ratio of 4a and 3a is slight increase, the yield of 4a significantly decreased. We had added this result in table 1 in the revised manuscript(see table 1, entry 15) .
- In Table 1, the authors ran the Ag2CO3 mediated reactionunder 60 degree Celsius, have the authors tried to lower the reaction temperature to see if the product ratio is still good or increased? Right now, the 60% Ag2CO3 is not a catalytic amount, have authors tried to lower the amount of Ag2CO3 under low reaction temperature (Such as room temperature or lower)?
Response: 4a is only obtained in 25% yield and with poor stereoselectivity when the reaction is carried out in the presence of 10 mol% of Ag2CO3 at room temperature for 72 h. We had added this result in table 1 in the revised manuscript (see table 1, entry 9) .
- Moderate editing of the English language is required. For example:
- i) In line 41-43, “Bhattacharjee and co-workers developed Ru complexes with different structures (i. e [Ru(dppe)(PPh3)(CH3CN)2Cl][BPh4] and [Ru(dppp)2 (CH3CN)Cl][BPh4]) catalyzed stereoselective N-vinylation of pyrazoles and alkynes…” Remove “with different structures”.
Response: We had corrected it in the revised manuscript.
- ii) In line 57-58, “In consideration of the importance of pyrazoles, we recently were interested in the synthesis of pyrazoles.” can be modified to “Considering the importance of pyrazoles, our recent research interests focused on the synthesis of pyrazoles.”
Response: We had corrected it in the revised manuscript.
iii) In line 83-84, “The reaction showed dependence on the reaction temperature, becoming more efficient with an increase in reaction temperature..” can be modified to “The reaction showed dependence on the reaction temperature, with increasing the reaction temperature, the reaction became more efficient.”
Response: We had corrected it in the revised manuscript.
- iv) In line 91-92, “Encouraged by this result, with the aim of increasing the yield of 4a, we studied the reaction in detail” can be modified to “Encouraged by this result, we further increased the amount of Ag2CO3.”
Response: We had corrected it in the revised manuscript.
- v) In line 95-96, “60oC seemed to be a suitable reaction temperature ofter the reaction temperature was checked” needs to be rewritten.
Response: In the revised manuscript, we have revised this statement(i.e., of the various reaction temperature checked, 60 oC was found to be optimal).
In addition, we have invited experts with good English to revise this manuscript, and English language and presentation are more easy and smooth.
Finally, we want to show our great respect to you and reviewers. Your critical reviews and suggestions definitely improved the quality of this manuscript.
Thank you once again for your efforts in this matter. I look forward to hearing from you.
Sincerely yours,
Haifeng Yu
